# Exploring the Influence of Social Class and Sex on Self-Reported Health: Insights from a Representative Population-Based Study

**DOI:** 10.3390/life14020184

**Published:** 2024-01-26

**Authors:** Luis Prieto

**Affiliations:** Distance Learning, Faculty of Public Health and Policy, London School of Hygiene and Tropical Medicine, London WC1E 7HT, UK; luis.prieto@lshtm.ac.uk

**Keywords:** health inequalities, self-reported health, social class, sex disparities, directed acyclic graphs (DAGs), cross-sectional study, European Health Survey, logistic regression, minimum adjustment set, socioeconomic factors

## Abstract

This study investigates the intricate interplay between social class, sex, and self-reported health (SRH) using data from the European Health Survey of Spain 2020 (EESE2020). Employing a cross-sectional design and a representative sample of 22,072 individuals, the analysis explores the persistence of disparities after adjusting for covariates, focusing on health-related variables. The study employs logistic regression models and directed acyclic graphs (DAGs) to delineate the direct effects of social class and sex on SRH, identifying a minimum adjustment set to control for confounding variables. Results reveal a gradient effect of social class on SRH, emphasizing the enduring impact of socioeconomic factors. Sex-based disparities in SRH diminish after considering additional health-related variables, highlighting the importance of a holistic approach. DAGs serve as transparent tools in disentangling complex relationships, guiding the identification of essential covariates. The study concludes that addressing health inequalities requires comprehensive strategies considering both individual health behaviours and socio-economic contexts. While recognizing limitations, such as the cross-sectional design, the findings contribute to a nuanced understanding of health disparities, informing evidence-based interventions and policies for a more equitable healthcare system.

## 1. Introduction

Health inequalities persist as a complex and pervasive challenge demanding rigorous scrutiny within the field of public health research [1]. The multifaceted nature of these disparities, cutting across diverse demographic and socio-economic strata, underscores the critical need for comprehensive investigations [2]. Beyond merely reflecting disparities in health outcomes, these inequalities highlight systemic and structural issues contributing to differential health experiences among various population groups [3]. Addressing these disparities is crucial, not only for promoting individual well-being but also for fostering a more equitable and just healthcare system [4]. A nuanced exploration of health inequalities is thus paramount to informing evidence-based interventions and policies aimed at mitigating these disparities and promoting health equity.

Building upon this recognition of persistent health inequalities, this investigation aims to address two key dimensions: sex-based disparities and social class differentials in self-rated health (SRH). Robust evidence underscores a sex-based inequality in SRH, revealing that women consistently report lower SRH compared to their male counterparts [5,6,7,8,9]. However, this observed sex disparity tends to diminish with the inclusion of other health status variables in the analytical framework [10,11], suggesting a dynamic relationship between sex, SRH, and their determinants. Simultaneously, social class emerges as another significant determinant of SRH [12,13,14,15], elucidating a discernible hierarchy wherein individuals from lower social strata exhibit compromised SRH. This complex interplay necessitates a thorough investigation into the intricate determinants of SRH inequalities, providing a foundation for targeted interventions to address the nuanced factors contributing to health disparities among diverse population groups.

Our study is meant to enrich the ongoing discourse surrounding the intricate relationship between sex/gender and SRH, building upon recent empirical research findings. Notable contributions from investigations such as Ryou et al. (2019) and Zeng et al. (2023) shed light on gender differences in the impact of SRH on mortality and emotional support’s association with SRH, respectively, among older adults [16,17]. Similarly, Park et al. (2020) and Akhtar et al. (2023) delved into the gender-specific dimensions of SRH, with the former exploring its connection to inflammation in Koreans and the latter revealing a clear gender gap in SRH among older adults in India, suggesting that women may be more sensitive to certain determinants of SRH [6,18,19]. Additionally, Vafaei et al. (2021) utilized intersectionality analysis to uncover the complex interplay of sex and social factors in shaping older adults’ perceptions of health in Canada [20]. Furthermore, Lysberg et al. (2021) explored age group changes in SRH over a 20-year period in Norway, noting a trending shift with a reduction in poor SRH among the youngest age group and an increase among middle-aged and older age groups, with women generally scoring lower than men [21]. In Eastern European countries, Gil-Lacruz et al. (2022) analyzed the gender gap in SRH from a generational perspective, revealing that individual characteristics, such as educational level or smoking, have a stronger effect on women’s perceived health than on men’s [22]. Lastly, Cui et al. (2021) investigated gender differences in the trajectories of SRH among Chinese older adults, finding no significant gender differences in the trajectories of SRH over time [23]. These diverse findings underscore the need for tailored health interventions that acknowledge the subtle ways in which sex and gender intersect with subjective health perceptions. Our study aims to contribute to this growing body of knowledge, examining sex-specific patterns in SRH within the context of a representative population-based study. The relationship between social class and self-rated health (SRH) has been a subject of extensive research as well, reflecting the multifaceted nature of health disparities. A comprehensive understanding of these disparities requires consideration of various aspects, including property ownership, authority, and credentials/skill, as suggested by the application of relational class theory to the United States by Eisenberg-Guyot & Prins (2020) [24]. A pilot study by McGarity-Shipley et al. (2023) investigating chronic shame as a potential mediator between subjective social status and SRH sheds light on the psychological pathways in middle-aged adults [25]. Furthermore, the English Longitudinal Study of Aging by Coustaury et al. (2023) emphasizes the importance of considering wealth, an often-neglected dimension, in understanding the association between subjective socioeconomic status and SRH [26]. In the Czech Republic, Hamplová et al. (2022) contributed to the concurrent validity of SRH by assessing the relative importance of physiological, mental, and socioeconomic factors [27]. Additionally, a study from Southwest China by Hu et al. (2021) explored health self-management as a mediator, revealing that lower social class predicts lower physical and mental health due to differences in health self-management abilities [28]. Trends in social class inequalities in disability and SRH among oldest old populations in Finland and Sweden, as studied by Enroth & Fors (2021), indicate increasing disparities over time [29]. In Spain, an intersectional analysis of gender, social class, and regional development by Pedrós Barnils et al. (2020) reveals both cumulative and heterogeneous SRH inequalities, emphasizing the joint contributions of material and psychosocial factors [30]. Lastly, a study by Lai et al. (2021) from Hong Kong demonstrates a social gradient of SRH in older people, highlighting the moderating role of the sense of community in the association between socioeconomic status and SRH [31]. This diverse array of studies underscores the need for a nuanced exploration of social class and its multifactorial impact on SRH across different contexts.

The objectives of this study include interrogating the persistence of these disparities after the adjustment for pertinent covariates, particularly self-reported health status variables. This inquiry begets a central question: What underlying determinants contribute to the association between social class and SRH? Moreover, how might these dynamics differ from patterns elucidated in gender-based health inequalities? The study aims to provide a comprehensive understanding of the nuanced factors contributing to health disparities, with a specific focus on the interaction between social class, sex, and SRH. Notably, the foundations of this inquiry rest upon a comprehensive and representative dataset sourced from the general population in Spain, extracted from the European Health Survey of Spain 2020 (EESE 2020) [32]. This dataset provides a rich source of information, enabling us to conduct in-depth analysis and draw meaningful insights into the multifaceted determinants of SRH across diverse social and demographic dimensions.

## 2. Materials and Methods

For this study, we employed a cross-sectional research design, utilizing data from the EESE 2020, a comprehensive survey conducted between 15 July 2019 and 24 July 2020 by the Ministry of Health and the Spanish National Institute of Statistics (INE) [32]. The EESE 2020, in its third edition, aimed to provide extensive health-related information for individuals aged 15 years and above in Spain, contributing to health planning and evaluation. Trained professionals conducted structured interviews to collect detailed information on a wide range of health-related topics, including demographic characteristics, lifestyle factors, preventive practices, and health outcomes.

The participants, selected by the INE using a three-stage stratified sampling methodology, included 22,072 individuals aged 15 years and older. Multi-stage sampling is a sampling technique used in survey research to select a sample from a large population [33]. It involves dividing the population into smaller subgroups called units and selecting a sample of units at each stage. This method is particularly useful for sampling large, geographically dispersed populations, as it can reduce the cost and time of data collection. This robust sampling strategy involved selecting census sections (municipalities), main family dwellings, and individuals to be surveyed, ensuring representation across diverse social and demographic dimensions. Initially, the primary sampling unit was identified as census sections, meticulously stratified based on the size of the municipality. These sections were then chosen with a probability proportional to their size, gauged by the number of main family dwellings they encompassed. In the subsequent stage, main family dwellings were systematically selected within each chosen section, guaranteeing an equal probability of selection. Finally, to obtain individual responses, an adult (aged 15 or older) was randomly chosen within each household for the interview. The overall sample size amounted to approximately 37,500 households, strategically distributed across 2500 census sections, with an average of 15 households selected per section. The stratification process involved categorizing municipalities based on their population size, ensuring the formation of a diverse and nationally representative sample. The analysis of the recovery rate for the survey, a crucial metric assessing the effectiveness of data collection, revealed that at the national level, the effective sample represented almost 59% of the theoretical sample, signifying that nearly 59% of the total households in the theoretical sample were successfully surveyed. Drilling down to the regional level, it was observed that most autonomous communities exhibited effective sample percentages ranging between 50% and 73%. Additional details regarding the multi-stage sampling technique and the survey methodology are available elsewhere [34]. It is important to underscore that the comprehensive approach employed and the unwavering adherence to established sampling procedures significantly contributed to the robustness and reliability of the survey data.

Approval from an accredited ethics committee was not required, as the EESE 2020 data were considered non-confidential, obtained from public and anonymous files. Informed consent was not necessary as the data were obtained from public and anonymous files. Participants selected for the EESE 2020 were informed by letter about their inclusion in the survey, the confidential nature of data collection, and the regulations protecting them.

For the study we employed a comprehensive set of variables derived from the survey, capturing various dimensions of health and well-being:Sex: Biological sex; distinguished between male and female participants. No information on gender identity, sexual orientation, LGBTQ, or X-gender individuals was captured.Age: Represented the age of the participants in years.Social Class: Initially classified participants into six categories [35]: Class I (directors and managers of establishments with 10 or more employees and professionals traditionally associated with university degrees), Class II (directors and managers of establishments with fewer than 10 employees, professionals traditionally associated with university degrees and other technical support professionals, and sportsmen and sportswomen), Class III (intermediate occupations and self-employed workers), Class IV (supervisors and workers in skilled technical occupations), Class V (skilled workers in the primary sector and other semi-skilled workers), and Class VI (unskilled workers). To enhance interpretability and streamline the analytical approach, a recoding strategy was employed, collapsing the six original categories into three broader classes: High (Classes I and II), Middle (Classes III and IV), and Low (Classes V and VI). This strategic adjustment aligns with the flexibility endorsed by the Spanish Society of Epidemiology, which recognizes alternative groupings in social class categorization [35]. Moreover, our decision is reinforced by recent research within the same dataset, where the three-category approach was consistently utilized to characterize social class while investigating factors influencing screening test uptake for colorectal cancer [36]. Also, the consistency observed in contemporary epidemiological studies across diverse datasets in Spain further support our rationale [37,38]. This approach not only ensures methodological alignment but also enhances the applicability and comparability of our findings within the broader epidemiological context in Spain.Chronic Conditions: Indicated the presence or absence of any chronic condition.Health Issues (last 12 months): A dichotomous variable capturing the occurrence of any of the 32 health conditions originally included in the survey. Participants responded to specific questions related to various health issues such as high blood pressure, heart attack, angina, arthritis, allergies, mental health conditions, and others over the last 12 months. Each health condition was initially coded separately, resulting in a set of binary variables indicating the presence or absence of each specific condition. The final variable was derived by summing the binary indicators for all 32 conditions. It serves as a dichotomous measure, classifying participants as either having experienced one or more health conditions or having no reported health conditions over the last 12 months. This consolidated variable simplifies the representation of the complex array of health issues, facilitating a comprehensive analysis of the overall health burden within the study population.Health Limitation (≥6 Months): Distinguished between individuals with or without health limitations.Pain (last 4 weeks): Categorized as “None”, “Very mild/mild”, “Moderate” or “Severe/Extreme”.Medicines (last 2 weeks): Indicated the use or non-use of medicines.Hospitalization (last 12 months): Distinguished between those who were or were not hospitalized.Body Mass Index (BMI): Categorized participants as ”Normal/Underweight”, ”Overweight”, or ”Obese”.Depression (last 12 months): Indicated the presence or absence of depression.Self-Reported Health (last 12 months): A binary variable representing ”Good/Very good” or ”Fair/Poor/Very poor” health perceptions.

The selection of these variables was guided by a conceptual model, as depicted in Figure 1, with the intention to determine the factors that contribute to SRH. In the figure, we employed a directed acyclic graph (DAG) to visually depict and analyse the relationships between the study variables [39,40,41,42]. The primary purpose of the DAG was to offer a conceptual model for investigating the complex interactions within the dataset, aiding in causal thinking. In the figure, the rectangular boxes represent DAG nodes, each corresponding to a specific variable, and arrows between nodes indicate hypothesized directional relationships. Notably, the graph is acyclic, ensuring a clear and non-circular representation of variable relationships. Within this conceptual model, the presence of an oval-shaped node signifies a latent variable (denoted as “Unknown”). This latent variable implies the potential influence of unobservable or unmeasured factors on health issues, chronic conditions, depression, and BMI. 

The complex web of relationships among the selected study variables is grounded in existing literature. The arrow from Sex to Age signifies well-established connections between biological sex and the aging process [43], while the arrow to Social Class is informed by studies highlighting sex and gender-based disparities in socio-economic status [44]. Sex’s association with health issues, chronic conditions, and depression reflects a body of research demonstrating sex-specific health disparities, where women often report different health outcomes than men [45,46,47,48]. Age’s relationship with health issues, chronic conditions, depression, and social class aligns with extensive literature showcasing the impact of age on health status and its interaction with socio-economic factors [49,50,51,52]. Social Class’s association with health issues, chronic conditions, depression, and obesity reflects a robust body of research demonstrating the profound impact of socio-economic factors on various health outcomes [53,54,55,56]. Obesity, in turn, is linked to health issues, chronic conditions, and depression [57,58,59,60,61]. Health issues and chronic conditions are central nodes influencing health limitations, pain, medicines, hospitalization, and depression [62,63,64,65,66,67,68,69,70]. Depression, in its relationship with health limitations, pain, medicines, and hospitalization, further underscores its pervasive influence on overall health [71,72,73,74,75]. Finally, the arrows from health limitations, pain, medicines, and hospitalization collectively shape individuals’ self-reported health, emphasizing the comprehensive nature of health-related determinants in subjective health assessments [76,77,78,79]. These connections, rooted in established literature, provide a solid theoretical foundation for our DAG.

We utilized descriptive statistics to summarize the demographic characteristics and prevalence rates of the variables in the study. We conducted logistic regression analyses to assess the odds ratios (OR) and 95% confidence intervals (CI) for each predictor variable. With these analyses, we aimed to reveal the individual impact of each predictor on the likelihood of reporting levels of self-rated health.

To scrutinize the complex interplay of variables within the conceptual model in Figure 1, we conducted a multiple logistic regression analysis. The model incorporated all variables, including sex, age, social class, chronic conditions, health issues, health limitations, pain, medicines, hospitalization, BMI, and depression. The logistic regression provided estimates of the associations between these variables and self-reported health, allowing for a comprehensive exploration of the determinants of subjective health assessments. We assessed the contribution of each variable in explaining the variance in self-reported health, yielding coefficients, standard errors, Wald statistics, and significance levels. Importantly, we employed the Hosmer and Lemeshow goodness-of-fit test to evaluate the Figure 1 model’s fit to the data, ensuring its appropriateness. Additionally, we calculated the Nagelkerke R Square, which is commonly used in logistic regression to assess the goodness-of-fit of the model. Unlike the R-squared in linear regression, Nagelkerke R Square does not measure the proportion of variation explained in the model in the same manner, as chi-square units are being assessed rather than linear sums of squares for a continuous dependent variable. Instead, it provides a measure of the improvement in model fit over a null (intercept-only) model, with higher values indicating better model fit. This statistic is useful for evaluating the overall explanatory power of the logistic regression model in capturing the relationship between the predictors and the binary outcome.

We used SPSS version 28 [80] for statistical analyses, encompassing descriptive statistics and logistic regression models. 

Utilizing DAGs in our study was pivotal for disentangling the intricate relationships within the data. DAGs function as graphical tools that aid researchers in visually representing and understanding the complex interplay of variables [41]. One of the primary purposes of employing DAGs in our investigation was to identify crucial variables that needed to be controlled for to obtain precise estimates of the direct effects of independent variables on the outcome [81]. The concept of “direct effect” is central to DAGs, denoting the unmediated influence that an independent variable exerts on the outcome. This is vital for isolating and comprehending the specific impact of variables of interest without the confounding influence of other variables.

Figure 2 provides a visual representation of these methodological concepts. The grey rectangular boxes highlight the minimum set of variables requiring control for accurate estimation, including health limitations, pain, medicines, and hospitalization. These variables were selected based on the application of a web-based tool, DAGitty [40]. DAGitty employs principles from causal inference and graph theory to construct DAGs that represent the relationships between variables [40]. In our study, each variable is represented as a node in the graph, and arrows between nodes indicate causal relationships. The acyclic nature of the graph ensures a clear direction of causation. The tool systematically identifies the minimal adjustment set (MAS) by assessing the graphical structure. The MAS consists of a minimal set of variables that need to be controlled for to estimate the direct effect of a particular variable on the outcome without introducing bias from confounding factors [81]. To illustrate the concept mathematically, let’s consider a causal relationship between two variables, A and B, where A influences B. If we want to estimate the direct effect of A on the outcome B, DAGitty helps identify the minimal set of variables (C, D, E, etc.) that, when controlled for, ensures an unbiased estimation of this direct effect. This is achieved by blocking all backdoor paths from A to the B, where a backdoor path is any path that ends with an arrow pointing into A. DAGitty’s mathematical algorithms are designed to systematically identify and present this minimal adjustment set, ensuring a robust and unbiased estimation of causal effects in observational studies.

Transitioning from this methodological groundwork, we applied a logistic regression model to examine the associations between the identified minimum set of variables and SRH, focusing on the direct effects. As in the full model, we used standard goodness-of-fit statistics to evaluate model performance. This minimal approach ensured a robust and consistent methodological approach for analysing the essential determinants of SRH, providing valuable insights without unnecessary complexity.

## 3. Results

Table 1 presents the demographic characteristics of the study participants, providing a comprehensive overview of key variables. Notably, the sample of 22,072 individuals exhibited a balanced distribution of sex, with 52.9% being female and 47.1%, male. Regarding age, the mean was 54.6 years (SD = 19), reflecting a diverse age ranging from 15 to 104 years. The social class distribution revealed representation across High (18.9%), Middle (34.5%), and Low (46.6%) categories. Notably, 70.6% of participants reported their SRH as Very good or Good, while 29.4% rated it as Fair, Poor, or Very poor. These baseline characteristics set the stage for exploring the relationships between sex, social class, and SRH in subsequent analyses. 

Table 2 provides a detailed examination of the association between various independent variables and participants’ SRH. The influence of sex on SRH is evident, with females being 53% more likely than males to report Fair, Poor, or Very poor health (OR = 1.53, 95% CI: 1.44–1.62). Age exhibited a modest impact, with each additional year being associated with 5% higher odds of reporting poorer SRH (OR = 1.05, 95% CI: 1.048–1.052). Social class revealed a gradient effect, as individuals in the Low social class category showed 153% higher odds of reporting poorer SRH compared to the High social class (OR = 2.53, 95% CI: 2.31–2.78). Chronic conditions, health issues, health limitations, pain, medicines, hospitalization, higher BMI, and depression all exhibited substantial associations with SRH. Notably, participants with severe limitations faced markedly increased odds of reporting poorer SRH.

Table 3 presents the outcomes of the full multiple logistic regression model, aligning with the comprehensive conceptual framework illustrated in Figure 1. Building upon the significant bi-variate associations identified in Table 2, which highlighted the initial impact of socio-demographic variables on SRH, the full model provides a better understanding of these relationships. Initially, both sex and social class exhibited noteworthy associations with SRH. Females were 53% more likely than men to report Fair, Poor, or Very poor health, while social class displayed a clear gradient effect, indicating higher odds of poorer health in lower classes. However, upon consideration of the full model, which accounts for the spectrum independent variables simultaneously, a compelling shift is observed. Controlling for age, chronic conditions, health issues, health limitations, pain, medicines, hospitalization, BMI, and depression, the association between sex and SRH disappeared (OR = 0.97, 95% CI: 0.89–1.06). This suggests that the initial disparities in SRH between males and females can be largely explained by these additional factors. In contrast, social class retained a robust association with SRH even after accounting for these variables, underscoring the persistent impact of socioeconomic disparities on health outcomes within the studied population. 

The model in Table 3 demonstrated satisfactory goodness-of-fit with a Nagelkerke R Square of 0.58. The Hosmer and Lemeshow Test yielded a non-significant result (Chi-square = 14.96, *p* = 0.06), supporting the model’s adequacy in fitting the observed data. Additionally, the observed-predicted classification of SRH achieved an overall percentage correct of 85.5%, affirming the model’s reliability in accurately predicting subjective health assessments within the study population.

Table 4 presents the results of the multiple logistic regression model employing the minimal sufficient adjustment set that aligns with the DAG in Figure 2. This model focused on the direct effects of sex and social class on SRH, providing a nuanced exploration while maintaining simplicity. In this minimal model, sex showed a negligible association with SRH (OR = 0.99, 95% CI: 0.91–1.07), indicating that, after accounting for health limitations, pain, medicines, and hospitalization, the initial sex-based disparities in SRH diminished substantially. However, social class retained a robust association with SRH, with individuals in the Low social class exhibiting 1.99 times higher odds of reporting poorer SRH compared to the High social class (95% CI: 1.77–2.25). This suggests that even in the absence of mediation by health limitations, pain, medicines, and hospitalization, social class remained a significant determinant of SRH. The impact of health limitations is also notable in this minimal model, with individuals experiencing severe limitations having 23.54 times higher odds of reporting poorer SRH (95% CI: 19.00–29.17). Pain, medicines, and hospitalization also maintained strong associations with SRH, reinforcing their independent contributions to subjective health assessments.

Comparing these results with Table 3, which represents the full multiple logistic regression model, we observe that the minimal model captured the essential associations without unnecessary complexity. Nagelkerke’s R Square (0.55) was consistent with the full model. Additionally, the observed-predicted classification of SRH achieved an overall percentage correct of 85.1%, demonstrating the reliability of the minimal model in predicting subjective health assessments.

## 4. Discussion

The intricate interplay between social class, sex, and SRH uncovered in this study prompts a more in-depth examination of the underlying mechanisms driving health inequalities. 

Social class, a key determinant of SRH, exhibits a gradient effect where individuals in lower classes consistently report compromised health compared to their higher-class counterparts. This gradient, observed even after accounting for a range of health-related variables, emphasizes the enduring impact of socioeconomic factors on subjective health assessments. The findings resonate with existing literature on social determinants of health [4,28,82,83], reinforcing the notion that addressing health inequalities requires comprehensive strategies that extend beyond individual health behaviours [5].

The disappearance of sex-based disparities in SRH when considering additional health-related variables challenges simplistic interpretations of sex inequalities in health outcomes [84]. The initial observation of women reporting lower SRH compared to men is nuanced by the inclusion of variables such as chronic conditions, health issues, and mental health, suggesting that these factors contribute significantly to the observed sex-based differences. This underscores the importance of adopting a holistic approach to understanding health inequalities, acknowledging the complex web of factors influencing subjective health assessments [85].

Social class, identified here as a pivotal determinant of SRH, presents a compelling case for targeted policy interventions [86]. Policymakers should strategically prioritize initiatives beyond conventional healthcare measures. By addressing systemic issues like education, employment, and housing inequalities, policies can actively uplift individuals in lower social classes [87]. A robust public health agenda must emphasize cross-sector collaboration and the seamless integration of social determinants into overarching policy frameworks. This multifaceted approach is essential to effectively narrowing the health gap.

Efforts towards social change should extend beyond a top-down approach. Community-driven health initiatives play a pivotal role in fostering lasting transformations [88,89]. Policymakers are urged to invest in empowering programs that enable communities to take an active role in enhancing their health. Whether through educational campaigns or job training initiatives, fostering social support networks can yield enduring positive outcomes. This grassroots involvement is integral to creating a resilient foundation for improved health on a community level.

In tandem with policy and community efforts, healthcare practitioners bear a responsibility to integrate social determinants into patient care. Routine screening for social determinants, including socioeconomic status, can serve as a compass for personalized interventions [90]. This approach acknowledges and addresses the broader contextual factors influencing health outcomes. By recognizing and acting upon the intricate links between social determinants and health, healthcare professionals contribute to a holistic model of care that goes beyond traditional biomedical perspectives.

This recognition of the intricate links between social determinants and health outcomes can stimulate collective action, prompting a ripple effect that transforms not only individual behaviours but also societal structures.

DAGs play a pivotal role in unravelling the complexity of these relationships. By visually depicting hypothesized directional relationships between key variables, DAGs guide researchers in identifying essential covariates necessary for unbiased estimates of specific independent variables’ direct impacts on the outcome [81]. The application of DAGs in this study facilitated a transparent and meticulous approach to modelling, ensuring that the identified minimum set of covariates captured the true direct effects of social class and sex on SRH. 

In this respect, we would like to elaborate on the rationale behind selecting the reduced model in Table 4 over the more comprehensive model in Table 3. While it is true that the larger model in Table 3 has significant effects for most predictors and exhibits better fitting statistics, our decision to emphasize the reduced model was driven by consideration of the interpretability of a simpler model and the principle of Occam’s razor (i.e., the idea that, all else being equal, simpler explanations are generally better than more complex ones). The goal of adopting the reduced model was to distil the essence of the relationships under investigation. The larger model, albeit statistically sound, may introduce unnecessary complexity, potentially clouding the direct associations we sought to highlight. The application of DAGs was instrumental in this simplification, guiding us to transparently and meticulously model the relationships. By choosing the reduced model, we aimed to offer a more straightforward and interpretable representation of the key factors directly impacting SRH. This strategic simplification, guided by DAGs, allows for clearer insights into the critical associations between social class, sex, and SRH.

While structural equation modeling (SEM) [91] could be a suitable approach for estimating the complex interrelationships depicted in our DAG, we opted for multiple logistic regression for several reasons. Our study focused on examining the associations between a range of socio-demographic factors and self-reported health, aiming for a more straightforward and interpretable model. Multiple logistic regression allowed us to assess the impact of various variables on the odds of reporting different health states, providing a clear and concise presentation of our findings. However, we acknowledge the robust analytical capabilities of SEM, particularly in capturing latent constructs and intricate pathways. Future analyses employing SEM could offer a more comprehensive exploration of the structural relationships implied by our DAG, providing additional insights into the subtle interactions among the studied variables.

In the broader context of research exploring the intersections of socio-demographic factors with SRH, our study makes a significant contribution by delving into the intricate dynamics of sex and social class. The extensive body of empirical research, as highlighted in the introduction, underscores the complexity and variability of these relationships across diverse populations. Our findings, derived from a representative population-based study, add to this rich tapestry by providing novel insights. We would like to highlight the cumulative nature of our contribution to the field. As future research endeavors unfold, it would be advantageous for researchers to consider the insights generated by our study as a foundation for designing more comprehensive and contextually informed investigations into the socio-demographic determinants of SRH.

Despite these contributions, it is crucial to recognize the study’s limitations. The cross-sectional design restricted our ability to establish causal relationships, and the reliance on self-reported data introduced the possibility of response bias. While the minimal sufficient adjustment set model offers a focused exploration, there may still be unmeasured confounders influencing the observed associations. 

We chose to represent age as a continuous variable in our logistic regression model, a decision made for both simplicity and precision in capturing the overall trend in its association with SRH. This methodological choice aligns with common epidemiological practices and enhances the interpretability of the results. However, it is crucial to acknowledge that age, when treated as a continuous regressor, assumes a linear relationship with SRH. The assumption of linearity suggests that the effect of each additional year is constant. Nevertheless, this simplicity may not fully encapsulate a more complex nature of the age–SRH relationship, and it is a point of legitimate concern raised by a reviewer.

Alternatively, treating age as a categorical variable in a multiple logistic regression model presents another valid avenue. This approach allows for a more flexible representation of the age–SRH relationship, capturing potential non-linearities and variations across different age groups. Categorizing age could provide a more detailed understanding of how subjective health assessments vary at different life stages. While our study provides valuable insights, future research endeavours can explore these alternative modelling approaches. 

To deepen the understanding of health determinants, future research should adopt a longitudinal approach to assess the sustained impact of policies aimed at reducing health inequalities. Longitudinal studies tracking changes in social class disparities and health outcomes over time can offer valuable insights into the effectiveness of interventions [92,93]. 

Moreover, exploring the intersectionality of various socio-demographic factors, including age, sex, race, ethnicity, and geographical location, would provide a more nuanced perspective [50,94]. Understanding how these factors intersect with social class can inform targeted and inclusive policies, fostering a comprehensive comprehension of health determinants.

Given the potential influence of unmeasured variables on the observed social class effect, evaluating future survey designs becomes imperative. Just as the impact of sex disparities diminishes after controlling for specific health-related variables, a more targeted approach to data collection may unveil a more detailed picture of social class determinants. Future surveys could incorporate additional indicators related to socioeconomic status, such as access to education, employment stability, and housing conditions. These targeted questions would allow for a more comprehensive examination of the multifaceted nature of social class and its intricate relationship with health outcomes [95,96,97]. Such refined measures have the potential to dissect the social class effect, revealing specific dimensions that significantly contribute to health inequalities.

## 5. Conclusions

The study highlights a clear gradient effect of social class on self-reported health (SRH), emphasizing the enduring impact of socioeconomic factors on health outcomes.

Initially, women reported lower SRH than men, but this disparity diminished after accounting for health-related variables, revealing the nuanced nature of gender-based health inequalities.

The application of directed acyclic graphs (DAGs) provided a methodologically robust approach, elucidating the minimum set of variables necessary for unbiased estimation of direct effects.

The findings underscore the need for comprehensive interventions addressing both individual health behaviours and broader socioeconomic determinants to tackle health inequalities.

## Figures and Tables

**Figure 1 life-14-00184-f001:**
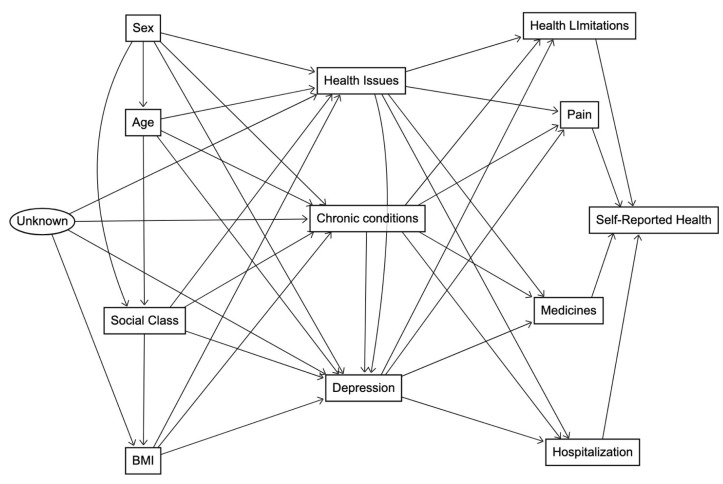
Directed acyclic graph (DAG) showing the relationships between study variables.

**Figure 2 life-14-00184-f002:**
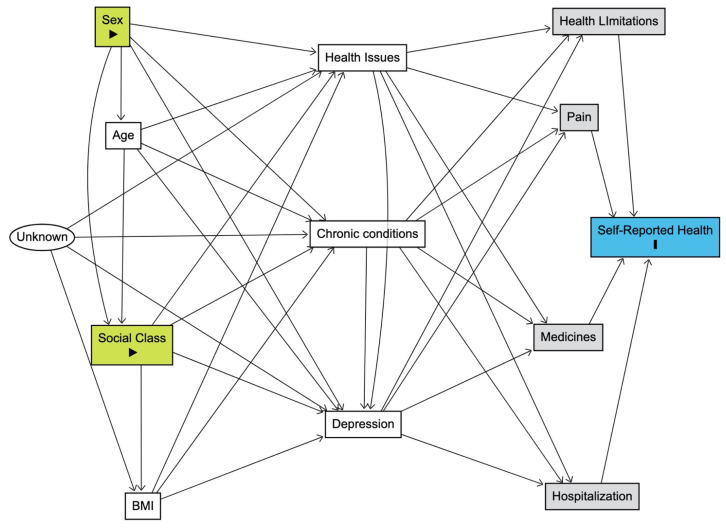
Directed acyclic graph (DAG) showing the minimal adjustment set (MAS) of variables that need to be controlled for to estimate the direct effects of sex and social class on self-reported health.

**Table 1 life-14-00184-t001:** Demographic characteristics of participants (*n* = 22,072 ^1^).

Sex	
Male	47.1%
Female	52.9%
Age	
Range, years	15–104
Mean (SD)	54.6 (19)
Social Class ^2^	
High	18.9%
Middle	34.5%
Low	46.6%
Chronic Conditions ^3^	
No	39.1%
Yes	60.9%
Health Issues (last 12 months) ^4^	
No	34.6%
Yes	65.4%
Health Limitation (≥6 Months) ^5^	
Not limited	72.3%
Limited	21.9%
Severely limited	5.8%
Pain (last 4 weeks)	
None	56.0%
Very mild, Mild	21.9%
Moderate	14.9%
Severe, Extreme	7.2%
Medicines (last 2 weeks) ^6^	
No	41.3%
Yes	58.7%
Hospitalization (last 12 months) ^7^	
No	91.8%
Yes	8.2%
Body Mass Index	
<25 (normal, underweight)	44.8%
25–29.9 (overweight)	39.0%
≥30 (obese)	16.2%
Depression (last 12 months)	
No	92.9%
Yes	7.1%
Self-Reported Health (last 12 months)	
Very good, Good	70.6%
Fair, Poor, Very poor	29.4%

^1^ Variables with missing values (%): Social class (4.5), Chronic conditions (0.1), Pain (0.1), BMI (5.3). ^2^ Based on occupation. ^3^ Chronic or long-term health condition or problem. ^4^ From a list of 32 health conditions. ^5^ Degree of limitation for at least 6 months due to health problems. ^6^ Prescribed by a doctor. ^7^ Hospital admission in the last 12 months, excluding birth or C-section.

**Table 2 life-14-00184-t002:** Association between each of the independent variables investigated and the self-reported health of participants.

Independent Variables	Self-Reported Health		OR (95% CI)
	Very Good, Good	Fair, Poor, Very Poor	
Sex			
Male	75.2%	24.8%	1
Female	66.5%	33.5%	1.53 (1.44–1.62)
Age			
Mean, years	50.07	65.40	1.05 (1.048–1.052)
Social Class			
High	82.5%	17.5%	1
Middle	73.2%	26.8%	1.73 (1.57–1.90)
Low	65.0%	35.0%	2.53 (2.31–2.78)
Chronic Conditions			
No	95.2%	4.8%	1
Yes	54.9%	45.1%	16.14 (14.55–17.91)
Health Issues			
No	95.3%	4.7%	1
Yes	57.6%	42.4%	14.91 (13.35–16.7)
Health Limitation			
Not limited	88.0%	12.0%	1
Limited	29.3%	70.7%	17.73 (16.4–19.18)
Severely limited	9.5%	90.5%	69.81 (57.54–84.70)
Pain			
None	87.6%	12.4%	1
Very mild, Mild	65.5%	34.5%	3.73 (3.45–4.04)
Moderate	38.6%	61.4%	11.27 (10.31–12.31)
Severe, Extreme	21.1%	78.9%	26.40 (23.15–30.12)
Medicines			
No	92.7%	7.3%	1
Yes	55.1%	44.9%	10.36 (9.50–11.29)
Hospitalization			
No	74.0%	26.0%	1
Yes	32.4%	67.6%	5.95 (5.37–6.60)
Body Mass Index			
<25 (normal, underweight)	78.3%	21.7%	1
25–29.9 (overweight)	69.3%	30.7%	1.59 (1.49–1.70)
≥30 (obese)	58.0%	42.4%	2.61 (2.40–2.84)
Depression			
No	74.3%	25.7%	1
Yes	22.9%	77.1%	9.71 (8.59–10.97)

**Table 3 life-14-00184-t003:** Multiple logistic regression model of the factors associated with self-reported health: full model.

Variables in the Model	B	S.E.	Wald	Sig.	Exp(B)	95% CI for EXP(B)	
						Lower	Upper
Sex							
Male (reference)							
Female	−0.03	0.046	0.50	0.481	0.97	0.89	1.06
Age (years)	0.02	0.001	119.64	<0.001	1.02	1.01	1.02
Social Class			102.03	<0.001			
High (reference)							
Middle	0.36	0.067	28.91	<0.001	1.43	1.26	1.63
Low	0.62	0.064	95.93	<0.001	1.87	1.65	2.12
Chronic Conditions							
No (reference)							
Yes	0.70	0.08	76.86	<0.001	2.01	1.72	2.35
Health Issues							
No (reference)							
Yes	0.55	0.085	41.19	<0.001	1.73	1.46	2.04
Health Limitations			1591.68	0			
Not limited (reference)							
Limited	1.74	0.049	1285.46	<0.001	5.70	5.18	6.27
Severely limited	2.76	0.118	550.51	<0.001	15.87	12.59	19.99
Pain			532.64	<0.001			
None (reference)							
Very mild, Mild	0.58	0.054	115.93	<0.001	1.78	1.60	1.98
Moderate	1.20	0.06	395.10	<0.001	3.32	2.95	3.74
Severe, Extreme	1.48	0.089	276.39	<0.001	4.41	3.70	5.25
Medicines							
No (reference)							
Yes	0.66	0.062	112.58	<0.001	1.94	1.71	2.19
Hospitalization							
No (reference)							
Yes	0.87	0.074	137.62	<0.001	2.39	2.06	2.76
BMI			19.75	<0.001			
Normal, underweight (reference)							
Overweight	0.09	0.051	3.32	0.068	1.10	0.99	1.21
Obese	0.28	0.062	19.72	<0.001	1.32	1.17	1.49
Depression							
No (reference)							
Yes	1.01	0.083	148.83	<0.001	2.74	2.33	3.22
Constant	−5.06	0.107	2242.80	0	0.06		

Model Summary: Nagelkerke R Square, 0.58. Hosmer and Lemeshow Test: Chi-square, 14.96, Sig. 0.06. Observed-predicted self-reported health, overall percentage correct: 85.5. NB: This table presents the outcomes of the full multiple logistic regression model aligned with the comprehensive conceptual framework illustrated in the DAG in Figure 1. The model incorporates all the variables in the DAG. The logistic regression provides estimates of the associations between the independent variables and self-reported health (SRH), allowing for a comprehensive exploration of the determinants of subjective health assessments.

**Table 4 life-14-00184-t004:** Multiple logistic regression model of the minimum number of factors associated with self-reported health: minimal sufficient adjustment set model.

Variables in the Model	B	S.E.	Wald	Sig.	Exp(B)	95% CI for EXP(B)	
						Lower	Upper
Sex							
Male (reference)							
Female	−0.01	0.043	0.09	0.765	0.99	0.91	1.07
Social Class			134.23	<0.001			
High (reference)							
Middle	0.41	0.065	41.04	<0.001	1.51	1.33	1.72
Low	0.69	0.061	126.88	<0.001	1.99	1.77	2.25
Health Limitations			2503.97	0			
Not limited (reference)							
Limited	2.04	0.045	2033.40	0	7.72	7.07	8.44
Severely limited	3.16	0.109	833.77	<0.001	23.54	19.00	29.17
Pain			736.83	<0.001			
None (reference)							
Very mild, Mild	0.69	0.051	184.17	<0.001	1.99	1.80	2.20
Moderate	1.34	0.057	549.40	<0.001	3.83	3.42	4.28
Severe, Extreme	1.62	0.084	371.88	<0.001	5.06	4.29	5.97
Medicines							
No (reference)							
Yes	1.36	0.052	683.47	<0.001	3.91	3.53	4.33
Hospitalization							
No (reference)							
Yes	0.91	0.071	164.81	<0.001	2.49	2.17	2.87
Constant	−3.78	0.073	2675.64	0	0.02		

Model Summary: Nagelkerke R Square, 0.55. Observed-predicted self-reported health, overall percentage correct: 85.1. NB: This table presents the results of the multiple logistic regression model employing the minimal sufficient adjustment set of factors described in the DAG in Figure 2. This model is focused on the direct effects of sex and social class on self-reported health, providing a nuanced exploration while maintaining simplicity.

## Data Availability

Publicly available datasets were analyzed in this study. This data can be found here: https://www.sanidad.gob.es/en/estadisticas/microdatos.do (accessed on 23 January 2024).

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
