# Peer review of "Exploring the Influence of Social Class and Sex on Self-Reported Health: Insights from a Representative Population-Based Study"

_life, 2024, doi:10.3390/life14020184_

Round 1
Reviewer 1 Report
Comments and Suggestions for Authors
I thought it was a very interesting study.
I would like to point out some points that should be clearly stated.
1. There is a great deal of empirical research on the relationship between gender and social class and SRH. Authors should clearly state what their new findings are. This should be explained in the introduction and clearly stated in the discussion.
2. The author deals with sex, but what kind of questions is the information based on in this survey?
Is it biological sex? Or is it sexual orientation, or should we understand it as gender identity? Are LGBTQ or X-gender people not included in the target audience? A clear explanation is required.
3. Please add information regarding the representativeness of this survey data. Regarding three-stage stratified sampling, could you please tell our readers more about multi-stage sampling?
Please organize and explain primary, secondary, and tertiary sampling units. Please also include the recovery rate.
4. About social class. The author has classified it into six categories, but please provide the basis for the categorization using literature.
5. Also, it says, "To enhance interpretability and streamline the analytical approach," "collapsing the six original categories into three broader classes," but considering the significance and purpose of this research, I think that the six categories are sufficient. I thought it was reasonable and meaningful. Conversely, by breaking down the categories, the amount of information decreases. I thought the resulting loss of knowledge was more serious. Authors should provide stronger and more persuasive reasons for categorization.
6. I think you need a basis for the DAG settings in Figure 1. This is because the DAG must have been set intentionally (including arbitrarily) by the author. For example, we need an academic basis for the Sex⇒Age arrow.
The model with the number of variables in this study can also be modeled based on previous research. It is possible to draw a DAG based on the author's arbitrariness without any theoretical background. However, testing an arbitrary model would have no academic significance. Authors are required to demonstrate in their paper that each path in the DAG is set based on evidence.
7. All the independent variables in Tables 3 and 4 seem to be categorical variables, but there is no information about the reference category. It should be made clear to the reader. In addition, I think it is necessary to enrich the table with a title and footnotes so that the results can be understood just by looking at the table.
Reviewer 2 Report
Comments and Suggestions for Authors
The manuscript is done well and addresses important issues of social justice and public health.
The document needs a much stronger discussion regarding implications of the findings for policy and practice. At least a couple of paragraphs would be needed to cover this important additional content that is meant to facilitate the broader impact of the article.
Also, readers knowledgeable in structural equation modeling (SEM) are very likely to wonder why SEM was not employed when the DAG pretty clearly suggests that such approach would be tailored to estimate the relevant model.
Further justification is needed for the selection of variables, beyond trusting to DAG logic. Why were these specific variable chosen and why were relationships expected to take a particular form?
A few other matters need to be addressed:
1. Referring to the analysis as "multivariate" is misleading. To statisticians and data analysts, "multivariate" implies the presence of multiple dependent variabels (which would be the case for SEM). Friendly advice is to refer to the model as "multivariable" or "multiple" logistic regression.
2. It is important to be clear that interpretation of model fit (R2) in logistic regression does not have the same meaning ass in least squares linear models. It is not quite corect to refer to Nagelkerke as measuring the proportion of variation explained in the model, because chi-square units are being assessed rather than linear sums of squres for a continuous dependent variable.
3. The interpretation of odds ratios does not quite seem correct. Saying, for example, "The influence of sex on SRH is evident, with 217 females demonstrating a 1.53 times higher odds of reporting Fair, Poor, or Very poor 218 health compared to males (OR = 1.53, 95% CI: 1.44-1.62)." is misleading. It would be ebtter to say that, in this context, that women have 1.53 times the odds of men of reporting that standard of health. It would be best to say that women are 53% more likely than men to report that health status; the current phrasing leads to the misleading conclusion that women are 153% more likely.
Reviewer 3 Report
Comments and Suggestions for Authors
The submitted paper entitled “Exploring the influence of social class and sex on self-reported health: insights from a representative population-based study” studies the crude and adjusted effects of sex and social class on a binary variable representing self-reported health (SRH), using a large dataset from the European Health Survey of Spain 2020 (22 072 subjects). As mentioned in the introduction, these effects have already been estimated in previous studies, and the results match our intuition that women consistently report lower SRH than men, and that individuals from the lowest social strata exhibit a more compromised SRH. The analyses in the present paper show: (a) a gradient effect of social class on SRH, emphasizing the enduring impact of 13 socioeconomic factors; (b) non-significant sex-based SRH disparities after considering additional health variables.
In order to obtain these findings, the author applies and describes a less common statistical approach. It is claimed that the combination of the two used methodologies provides a “simple” model that retains the information from another more “traditional” and complex regression model, and that the smaller model identifies the important health factors that override the sex/social class effects on SRH. This in fact would be the most interesting part of the paper, in my opinion. Unfortunately, I can’t see advantages in the smaller model chosen by the author and thus the paper loses its interest to me. Together with its epidemiological findings, that are not too sounding, I regret to say that my opinion is to reject the present paper.
In which follows, I substantiate my opinion.
1. The author mentions the Hosmer-Lemeshow goodness-of-fit test for regression analysis but he withdraws wrong conclusions from it, and fails to apply it correctly.
- Due to its definition, the test can only be applied to regression models with categorical predictors. The author incorrectly applies it to the model described in Table 3, which has Age as a continuous (not categorical) predictor. We cannot therefore rely on the conclusion he takes that the model has a good fit. The test was wrongly used.
- The test is also applied to the model in Table 4 (this time, correctly applied since there are no continuous predictors). and provides p<0.001. The author states that therefore the model has a good fit, which is wrong. It is precisely the contrary: we reject the null that the model has a good fit.
I recall that this second model, in Table 4, is the one the author emphasizes in his conclusions as the best model. And it does not have a good fit.
2. The author did not consider the inclusion of interaction terms between sex/social class and the remaining predictors. Why not? That would have told us if the effect of the predictors is different in men and women / in different social strata.
3. I don’t understand why the “reduced” model in Table 4 (if it were a model with a good fit) would be better than the more complete model in Table 3. Note that the model from Table 3 only has significant effects, except for Sex. Why shouldn’t the author consider them all? The sample size is large enough to accommodate a complete model. If the question is on the identification of the predictor that implies Sex to lose its significance, then a different approach could do the job – perhaps, and most likely, a model smaller than that of Table 4 would be found.
Moreover, the model in Table 3 has several confidence intervals for the OR estimates that are worse (wider) than those from the model in Table 4.
4. Age is a continuous variable and is treated in the paper as a continuous predictor. The estimated effect on SRH is the following: “each additional year associated with 1.05 times higher odds of reporting poorer SRH”. Although the interpretation is correct mathematically speaking, it is incorrect from an epidemiological point of view. Aging from 20 to 21 years-old is very different from aging from 75 to 76 years-old. Thus the odds associated with that “1 year increase” cannot be the same.
The author should have discretized the variable Age, or else used generalized additive models.
5. Nagelkerke R-square does not have the same meaning in logistic regression as R-squared does in linear regression. It is wrong to say that it gives the proportion of variance (of SRH) explained by the model. It does not have a simple interpretation
6. Prediction ability of the regression models: the value of the percentage of correct classification from the contingency table with the observed responses and the predicted classes is not enough. We also need to address sensitivity and specificity of the model. Usually, it performs less well in the less represented response class.
Round 2
Reviewer 2 Report
Comments and Suggestions for Authors
The manuscript has been revised exensively, largely responsive to reviewers' recommendations.
The findings seem reasonably clear.
Comments on the Quality of English LanguageEnglish usage seems fine, although the usual copyeditor's attention to detail is needed.
Reviewer 3 Report
Comments and Suggestions for Authors
To the author:
I found that the revised version is better argued and supported than the previous one. I also appreciated the author's answers.
It is well-known that a modelling process requires a balance between simplicity and detail. The author looks for a concise model that "retains" (loosing precision) the main conclusions from a more precise and detailed model. Why?! if there are no issues regarding sample size in the larger model, nor signs over fitting?! Moreover, the fitting in the larger model is better than the fitting in the smaller model. I still do not understand this, even after your answer.
That being said, the paper is well written and , in this second version, the author has reinforced the "importance" of the approached subject through adequate bibliographic references.
Precise remarks:
- The author is correct when stating the possibility of application of the Hosmer-Lemeshow test. I made a confusion with the necessary conditions of application for the deviance goodness-of-fit test.
-"In the revised manuscript, we have removed the reference to the Hosmer-Lemeshow test in Table 4." - still not removed. Please remove. I stress that the test suggests the fitting is not adequate.
- I am not convinced that the inclusion of a linear effect for age was a good choice. If age were to be considered a continuous regressor, then the effect would most likely have to be nonlinear. Thus, the studied "overall trend", mentioned by the author, might be too simple. (While agreeing that a categorization is less precise than the underlying continuous trait, it is the simplest roundabout to the inclusion of a nonlinear effect)
